# Hedgehog Signaling as a Therapeutic Target for Airway Remodeling and Inflammation in Allergic Asthma

**DOI:** 10.3390/cells11193016

**Published:** 2022-09-27

**Authors:** Anthony Tam, Emmanuel Twumasi Osei, Chung Y. Cheung, Michael Hughes, Chen X. Yang, Kelly M. McNagny, Delbert R. Dorscheid, Gurpreet K. Singhera, Teal S. Hallstrand, Stephanie Warner, James C. Hogg, Tillie L. Hackett, Chinten J. Lim, Don D. Sin

**Affiliations:** 1Centre for Heart Lung Innovation, St. Paul’s Hospital, Vancouver, BC V6Z 1Y6, Canada; 2Department of Pediatrics, University of British Columbia, Vancouver, BC V6H 0B3, Canada; 3Department of Biology, Okanagan Campus, University of British Columbia, Kelowna, BC V1V 1V7, Canada; 4School of Biomedical Engineering, University of British Columbia, Vancouver, BC V6T 1Z3, Canada; 5Division of Pulmonary, Critical Care, Sleep Medicine and the Center for Lung Biology, University of Washington, Seattle, WA 98195-6522, USA

**Keywords:** hedgehog signaling, asthma, TGFβ, airway remodeling, inflammation

## Abstract

Genome-wide association studies (GWAS) have shown that variants of patched homolog 1 (*PTCH1*) are associated with lung function abnormalities in the general population. It has also been shown that sonic hedgehog (SHH), an important ligand for PTCH1, is upregulated in the airway epithelium of patients with asthma and is suggested to be involved in airway remodeling. The contribution of hedgehog signaling to airway remodeling and inflammation in asthma is poorly described. To determine the biological role of hedgehog signaling-associated genes in asthma, gene silencing, over-expression, and pharmacologic inhibition studies were conducted after stimulating human airway epithelial cells or not with transforming growth factor β1 (TGFβ1), an important fibrotic mediator in asthmatic airway remodeling that also interacts with SHH pathway. TGFβ1 increased hedgehog-signaling-related gene expression including *SHH*, *GLI1* and *GLI2*. Knockdown of *PTCH1* or *SMO* with siRNA, or use of hedgehog signaling inhibitors, consistently attenuated *COL1A1* expression induced by TGFβ1 stimulation. In contrast, *Ptch1* over-expression augmented TGFβ1-induced an increase in *COL1A1* and *MMP2* gene expression. We also showed an increase in hedgehog-signaling-related gene expression in primary airway epithelial cells from controls and asthmatics at different stages of cellular differentiation. GANT61, an inhibitor of GLI1/2, attenuated TGFβ1-induced increase in COL1A1 protein expression in primary airway epithelial cells differentiated in air–liquid interface. Finally, to model airway tissue remodeling in vivo, C57BL/6 wildtype (WT) and *Ptch1^+/−^* mice were intranasally challenged with house dust mite (HDM) or phosphate-buffered saline (PBS) control. *Ptch1^+/−^* mice showed reduced sub-epithelial collagen expression and serum inflammatory proteins compared to WT mice in response to HDM challenge. In conclusion, TGFβ1-induced airway remodeling is partially mediated through the hedgehog signaling pathway via the PTCH1-SMO-GLI axis. The Hedgehog signaling pathway is a promising new potential therapeutic target to alleviate airway tissue remodeling in patients with allergic airways disease.

## 1. Introduction

Asthma is a complex disease that is defined by reversible airflow limitation, airway hyper-reactivity, remodeling and type 2 inflammation. Current data suggest that the airway epithelium plays a key role in modulating airway inflammation and remodeling in asthma. Although there is no cure for asthma, current medications are able to manage the inflammatory symptoms without affecting airway remodeling. One of the important epithelial mediators released to stimulate mesenchymal cell activation, increased expression and deposition of extracellular matrix (ECM) proteins such as collagen during airway remodeling in asthma is transforming growth factor-β (TGFβ) [1,2,3,4].

In addition to mediators such as TGF-β, recent genome-wide association studies (GWAS) have been used to identify potential molecular drivers of lung function decline which is caused by airway remodeling in asthma [5]. In a study by Li and colleagues, a subset of normal lung function genes associated with the hedgehog signaling pathway including the patched homolog 1 (PTCH1) and hedgehog interacting protein (HHIP) predicted lung function decline in white and African American subjects with asthma [6].

PTCH1 encodes a transmembrane protein receptor for the secreted hedgehog ligands including sonic hedgehog (SHH), Indian hedgehog (IHH) and desert hedgehog (DHH), which signal through smoothened (SMO) and a family of zinc-finger transcription factors called glioma-associated oncogene homologue (GLI1, 2, 3) [7,8]. After the activation of SMO, GLI transcription factors (GLI1, GLI2 and GLI3), which are a family of terminal effector proteins, can be activated via canonical and non-canonical hedgehog signaling pathways [8,9]. More specifically, the activation of GLI1 and GLI2 leads to the upregulation of genes that control cell proliferation, survival, angiogenesis and tumor growth [7,10]. Hedgehog signaling may be targeted with pathway inhibitors. Cyclopamine, an alkaloid isolated from *V. californicum* that binds strongly to the hepta-helical transmembrane domain of SMO, has been shown to inhibit SMO-mediated signaling [11,12]. Although cyclopamine is known to be very specific in targeting SMO in the hedgehog pathway, it is important to note that cyclopamine has been shown to induce tumor cell apoptosis independent of SMO [13]. GANT61 is a GLI antagonist with demonstrated specificity towards the disruption of the GLI1 and GLI2 DNA-binding ability [14].

Hedgehog signaling is activated in the airway epithelium of asthmatics. It is also up-regulated in the airways of mice exposed to house dust mite (HDM), which is a common stimulus to model allergic airways disease in mice [15]. Mice exposed to HDM demonstrate increased eosinophil counts and type 2 cytokine (IL4, IL5 and IL13) expression in bronchoalveolar lavage (BAL) fluid [16,17]. Treatment of purified eosinophils from HDM-exposed mice in culture with SHH induces IL4 expression, which illustrates a novel crosstalk mechanism between hedgehog signaling and asthma [18]. In contrast, intranasal administration of IL4 protein induces SHH protein expression in mice airway epithelium [19], which further supports this link.

Indeed, TGF-β was also shown to induce the epithelial–mesenchymal transition (EMT) in primary airway epithelial cells in patients with asthma [20], and there is also extensive evidence of crosstalk between hedgehog signaling and TGFβ response [14,21,22]. Mice exposed to HDM demonstrate airway tissue remodeling, which is confirmed by an increase in TGFβ protein secretion in the BAL and an increase in collagen, vimentin and α-smooth muscles in the sub-epithelial airway wall compartment [16]. However, the mechanistic crosstalk between hedgehog signaling and TGFβ1-mediated airway epithelial remodeling has not been thoroughly investigated.

The goal of this study was to investigate the biological role of PTCH1, SMO and GLI in airway epithelial cell models using various gain and loss of function assays and pharmacologic inhibition, and in a house dust mite model of allergic asthma using mice haplodeficient in *Ptch1* (*Ptch1*^+/−^). The results provide mechanistic insights in airway remodeling by demonstrating that interaction between TGF-β and hedgehog signaling is directly involved in ECM deposition in the airways which may have important implications for therapeutic studies in asthma. 

## 2. Material and Methods

### 2.1. Gain and Loss of Function Assays

Human airway epithelial cell line (1HAE_0_) derived from a male Caucasian donor was obtained from Dr. Dieter Gruenert University of California, San Francisco [23] and cultured in DMEM (Gibco BRL, Grand Island, NY, USA) with 10% fetal bovine serum (FBS). For the gene silencing study, cells were cultured in reduced serum (1% FBS) and supplemented with 10 ng/mL TGFβ1 (580704, BioLegend, San Diego, CA, USA) for 4 days followed by treatment with 10µM silencer-select scrambled siRNA (4390843; ThermoFisher Scientific, Waltham, MA, USA), *PTCH1* siRNA (4392420; ThermoFisher Scientific, Waltham, MA, USA) or *SMO* siRNA (s13164; ThermoFisher Scientific, Waltham, MA, USA) for an additional 72 h in the presence or absence of TGFβ1. The media-only control and TGFβ1 supplemented media were replaced 3 times a week.

To transiently over-express *Ptch1*, 1HAE_0_ cells were transfected with a plasmid encoding PTCH1-YFP (#58456 Addgene, Watertown, MA, USA), a fusion construct of mouse *Ptch1* with monomeric yellow fluorescent protein, using lipofectamine (L3000008, ThermoFisher Scientific, Waltham, MA, USA) in reduced serum (1% FBS) [24]. This was followed by 4 days of treatment with 10 ng/mL TGFβ1 in DMEM or media-only control which was replaced 3 times a week. 

For the intervention study, 1HAE_0_ cells were cultured in reduced serum (1% FBS) and pre-treated with 4.6 µM cyclopamine (SMO-smoothened inhibitor) (S1146; Selleckchem Houston, TX, USA), 50 µM GANT61 (GLI1/2 inhibitor) (S8075; Selleckchem, Houston, TX, USA) or ethanol vehicle for 3 days followed by 4 days of TGFβ1 treatment with fresh media and drugs replaced 3 times a week.

For the smoothened and GLI1/2 gene manipulation experiments, human primary airway epithelial cells (PAECs) from 3 healthy non-smokers cultured in air–liquid interface (ALI) for 21 days [25,26] were pre-treated with ethanol vehicle control (EtOH), 4.6 µM cyclopamine (SMO-smoothened inhibitor) (SS146, Selleckchem, Houston, TX, USA) or 50 µM GANT61 (GLI1/2 inhibitor) (S8075, Selleckchem, Houston, TX, USA) for 3 days followed by treatment of 10 ng/mL TGFβ1 for 7 days with fresh complete media containing drugs and TGFβ1 replaced 3 times per week.

### 2.2. Real-Time PCR

PCR methods and analysis have been previously described [26]. Real-time PCR was performed on a CFX384 Touch Real-Time PCR Detection System (Biorad, Hercules, CA, USA) using human TaqMan gene expression assays (ThermoFisher Scientific, Waltham, MA, USA) to measure *PTCH1* (Hs00181117_m1), *SMO* (Hs01090242_m1), *GLI1* (Hs00171790_m1), *GLI2* (Hs00257977_m1), *GLI3* (Hs00609233_m1), *SHH* (Hs00179843_m1), *CDH1* (Hs01023894_m1), *COL1A1* (Hs00164004_m1), *MMP2* (Hs01548727_m1) and *VIM* (Hs00958111_m1) gene expressions with normalization to *GAPDH* (Hs02758991_g1). Mouse TaqMan gene expression assay for *Ptch1* (Mm00436026_m1) was used to assay mouse *Ptch1* gene expression with normalization to human *GAPDH*. Gene expression was expressed as ΔΔct fold changes with normalization to control values, and Δct when multiple genes are displayed in parallel to show relative abundance.

### 2.3. RNA Sequencing of Airway Epithelial Cells from Asthmatic and Non-Asthmatic Subjects

Subject demographics and methods of sample collection, cell culture and RNA sequencing have been previously described [27]. In brief, human primary airway epithelial cells were obtained via endobronchial airway brushings from age-matched non-asthmatic healthy controls (*n* = 5) and asthmatic subjects (*n* = 10). All samples were obtained with informed consent, and the study was approved by the Providence Health Care Research Ethics board (H13–02173) at the University of British Columbia. All methods were carried out with relevant guidelines and regulations according to the Declaration of Helsinki. The primary airway epithelial cells were cultured in a bronchial epithelial growth medium (BEGM) (Lonza, Walkersville, MD, USA). At passage 2/3, these cells were grown at air-liquid-interface (ALI) on 0.4μm polyester transwell inserts (Corning, New York City, NY, USA) as described previously [28]. The samples were harvested on days 1, 5, 11 and 20 after confluent cells were transitioned to ALI. 

For both the RNA sequencing and RT-PCR experiments, total RNA samples were extracted from human airway epithelial cells and 1HAE_0_ cells using the RNeasy mini plus kit (74136; Qiagen, Hilden, DE) and reverse-transcribed into cDNA using the iScript cDNA synthesis kit according to the manufacturer’s protocol (1708891; Biorad, Hercules, CA, USA). Candidate gene RNA sequencing of primary epithelial cells was then performed on the Illumina Hiseq 2000 platform. For RNA-seq processing: The FASTQ reads were aligned to GENCODE GRCh37 (version 31) genome reference using STAR and quantified to gene counts using RSEM. The gene count was normalized to log_2_ counts per million (CPM) using limma voom. Genes with low expression were filtered out so that all the remaining genes have log_2_ CPM > 1 in at least 8 samples [29,30].

### 2.4. Immunofluorescence Microscopy on Differentiated Cell Culture

Methods for immunofluorescence staining of ALI cultured cells have been previously described [26]. In brief, the apical compartment was gently washed with PBS to remove excess mucous secretion and fixed with 10% phosphate-buffered formalin for 30 min at room temperature. Cells were permeabilized with 0.5% Triton X-100, which was followed by the application of a serum-free protein blocker (X090930–2, Agilent/DAKO, Santa Clara, CA, USA) for 1 h at room temperature. Membranes were washed with PBS 3 times for 5 min each, which was followed by incubation with a primary antibody against COL1A1 (39952S; Cell Signaling, Danvers, MA, USA) overnight at 4 °C. On the following day, membranes were washed with PBS, 3 times for 5 min each, which was followed by incubation with IRDye^®^ 800CW Goat anti-Mouse IgG (A-11005, ThermoFisher Scientific, Waltham, MA, USA) for 2 h at room temperature. Membranes were washed 3 times as described above, detached with a fine scalpel and placed with cells facing up on a microscopic glass slide after which cellular nuclei were counterstained with DAPI. The membrane was cover-slipped and imaged using confocal microscopy. COL1A1 protein intensity was quantified using Image J and normalized to the total number of DAPI-stained nuclei. 

### 2.5. Mice

Mice homozygous for the *Ptch1* mutation are embryonic lethal [31]; therefore, mice haplodeficient in *Ptch1* were established as previously described [26]. In brief, *Ptch1^tm1Mps^*/J (JAX 00308) mice were purchased from The Jackson Laboratory (Bar Harbor, ME, USA) and were maintained by heterozygous backcross to C57Bl/6J mice at the Biomedical Research Centre specific-pathogen free transgenic facility. Female mice of 10–16 weeks were used for these experiments. Experimental procedures were approved by the Animal Care Committee (Protocol #A15-0113) of the University of British Columbia based on guidelines provided by the Canadian Committee on Animal Care.

### 2.6. HDM Treatment Protocol

The house dust mite (HDM) treatment protocol has been previously described [26,32]. In brief, HDM extracts (endotoxin < 2500 EU/mg) (Greer Lab, Lenoir, NC, USA) were prepared in PBS after which 40 µL volumes were administered intranasally. Each aliquot consisted of 100 µg HDM extract per mouse for three consecutive days (days 0, 1 & 2) and then 20 µg/mouse for days 19, 20, 21 and 22. Twenty-four hours after the last exposure (day 23), lungs were lavaged, and blood and lung tissues were collected after sacrifice. 

### 2.7. Cell Differential and Mouse Lung Tissue Assessments

Cells from broncho-alveolar lavage (BAL) fluid were cytospun and stained with Wright’s stain for differential cell count. Neutrophils, macrophages, eosinophils, and lymphocytes were counted from at least 200 cells in a randomly selected field and expressed as a percentage of the total cell count. Masson’s Trichrome staining was performed to assess collagen expression (green color). The total intensity of the green color in the sub-epithelial wall region was normalized to basement membrane length using the Aperio imaging system. Immunohistochemical staining for Ki67 protein (ab16667; Abcam, Cambridge, UK) was performed on FFPE-mouse lung sections using a Bond polymer refined red detection kit on the Leica Bond Autostainer according to the manufacturer’s protocol. Ki67+ cells in the sub-epithelial wall region were normalized to the length of the basement membrane. 

### 2.8. Serum Cytokine Measurements

At the end of the experiment, blood was collected from the inferior vena cava of each mouse and placed in a 1.5 mL centrifuge tube to clot for 30 min at room temperature. Samples were subjected to centrifugation for 5 min at 12,000 rpm, aliquoted and stored at −80 °C. A mouse Magnetic Bead Luminex Panel (MTH17MAG-47K, Millipore) was used to determine mouse serum cytokines using the Luminex^TM^ 200^TM^ Instrument System (ThermoFisher Scientific, Waltham, MA, USA) following the manufacturer’s protocol.

### 2.9. Statistical Analyses

Data were tested for normality prior to the selection of a parametric (normal distribution) *t*-test or Mann–Whitney (non-normal distribution) test, one-way ANOVA with Bonferroni’s multiple comparisons test, or linear regression test, where appropriate. All data were analyzed with GraphPad Prism version 8 (GraphPad Software Inc., San Diego, CA, USA) and were expressed as mean ± SEM. Statistical significance was considered at *p* < 0.05.

## 3. Results

### 3.1. PTCH1 Knockdown Reduces TGFβ1-Induced COL1A1 and MMP2 Gene Expression in 1HAEo Cells

Previous studies of endobronchial biopsies showed that hedgehog signaling is activated in the airways of children with allergic asthma, and this evidence was confirmed by significant airway remodeling [15,33]. To determine the mechanistic link between hedgehog signaling and TGFβ1-mediated remodeling response, we stimulated human airway epithelial cell line 1HAE_o_ with TGFβ1. We showed that *SHH*, *GLI1* and *GLI2* gene expression levels were significantly increased after TGFβ1 treatment (Figure 1A). Since SHH has been reported to be an important ligand for PTCH1 receptor activation [7,8,13,14], we modulated PTCH1 expression via gene silencing and over-expression to determine its potential crosstalk with TGFβ1 signaling. We showed that treatment with PTCH1 siRNA alone reduced *PTCH1* gene expression by 60% but no differences were observed in *SMO*, *GLI1*, *GLI2*, *GLI3* and *SHH* gene expression levels compared to scrambled siRNA-treated controls, respectively (Figure 1B). We profiled the expression levels of *COL1A1* (collagen 1A1), *MMP2* (matrix metalloproteinase 2), and *CDH1* (epithelial cadherin), as these genes have been implicated in EMT [34]. *PTCH1* knockdown attenuated TGFβ1-induced increase in *COL1A1* and *MMP2* but did not modulate *CDH1* gene expression (Figure 1B–D). *Ptch1* over-expression significantly increased *Ptch1* gene expression but no differences were observed in *SMO*, *GLI1*, *GLI2* and *GLI3* gene expression when compared to untreated controls (Figure 2A). In contrast, *Ptch1* over-expression significantly augmented TGFβ1-induced increase in *COL1A1* and *MMP2* but did not affect *CDH1* gene expression (Figure 2B–D). In summary, gain and loss of function of *PTCH1* modulated TGFβ1-induced increase in *COL1A1* and *MMP2* gene expression in a human epithelial cell line.

### 3.2. TGFβ1-Induced COL1A1 Gene Expression Signals through the SMO-GLI1/2 Axes

To determine the role of SMO in canonical hedgehog signaling in response to TGFβ1, hedgehog-related genes were measured in 1HAE_o_ cells with and without siRNA-mediated knockdown of *SMO*. *SMO* gene expression was reduced in cells treated with *SMO* siRNA when compared to scrambled siRNA-treated controls, while expression of *PTCH1*, *GLI1*, *GLI3* and *SHH* was unchanged (Figure 2A). Knockdown of *SMO* significantly reduced TGFβ1-induced increase in *COL1A1* gene expression, but not of *MMP2* and *CDH1* (Figure 3A–C). Pre-treatment of cells with cyclopamine (a SMO inhibitor), and GANT61 (a GLI1/2 inhibitor) attenuated TGFβ1-induced increase in *COL1A1* gene expression (Figure 3D). Pre-treatment of cells with GANT61 but not cyclopamine significantly attenuated TGFβ1-induced increase in *MMP2* gene expression (Figure 3E). TGFβ1 reduced *CDH1* gene expression to a similar extent in the presence or absence of cyclopamine or GANT61 (Figure 3F). 

Next, to demonstrate the contribution of hedgehog signaling on TGFβ1-induced remodeling in primary airway epithelial cells differentiated in air–liquid interface (ALI), we treated ALI cultures with cyclopamine and GANT61, and showed that pharmacological targeting of the SMO-GLI axis attenuated TGFβ1-induced increase in COL1A1 protein expression (Figure 4A,B). Collectively, we showed that TGFβ1-induced increase in COL1A1 expression is partially mediated through the SMO-GLI axis of the canonical hedgehog signaling pathway.

### 3.3. Primary Airway Epithelial Cells Exhibit Upregulation of Hedgehog Signaling

Airway epithelial cells cultured at air–liquid interface (ALI) will differentiate from a basal monolayer into a pseudostratified epithelium. To model the temporal expression of hedgehog-signaling-related genes at different time points of epithelial differentiation, we characterized the gene expression of *SHH*, *SMO* and *GLI2*. Levels of *SHH*, *SMO* and *GLI2* were significantly increased on day 5, 11 and 20 post-ALI compared to day 1 post-ALI in both asthmatics and non-asthmatics (Figure 5A,B,D). There was no difference in *SHH*, *SMO*, *GLI2*, *GLI3* and *PTCH1* gene expression between non-asthmatic and asthmatic subjects at each of the collected time points (Figure 5A–F). Collectively, our primary cell data showed an upregulation of SHH, SMO, GLI2 and PTCH1 gene expression with epithelial differentiation independent of a diagnosis of asthma.

### 3.4. Ptch1^+/−^ Mice Fail to Induce Total Airway Wall Collagen Protein Expression in Response to HDM Exposure

Our in vitro mechanistic data prompted us to determine whether hedgehog signaling may be involved in airway remodeling in vivo. To evaluate the role of *Ptch1* in a mouse model, we turned to a previously described airway remodeling induction experiment using house dust mite (HDM) as a proof-of-concept [32,35,36]. Because homozygous *Ptch1*^−/−^ mice are embryonic lethal, we compared *Ptch1*^+/−^ mice and wildtype (WT) littermates challenged with repeated exposure to HDM (or a PBS vehicle control). The mice were then sacrificed and their lung sections were stained with Trichrome’s stain and Ki67 (Figure 6A,B). After exposure to HDM, *Ptch1^+/−^* mice demonstrated a significant decrease in Ki67+ cells in the sub-epithelial compartment but not in total collagen protein expression compared to WT mice (Figure 6C,D). We observed a significant positive correlation between Ki67+ cells and collagen expression in the sub-epithelial compartment (Figure 6E). Although collagen content is dependent on finely tuned turnover pathways, our results suggest that Ptch1 could play a critical role in sub-epithelial collagen expression in response to HDM exposure.

### 3.5. Ptch1^+/−^ Mice Fail to Induce Il-4 and Il-5 Serum Protein Expression in Response to HDM Exposure

To determine whether *Ptch1* plays any modulatory role in immune responses when challenged with HDM, differential cell counts from BAL fluid were evaluated using a Wright’s stain (Figure 7A). Total cell counts were increased in WT and *Ptch1^+/−^* mice to a similar extent after HDM exposure compared to their respective controls (Figure 7B). In a randomly selected field of view of cytospun BAL sample, the relative proportion of macrophages expressed as a percentage was displaced by increases in neutrophil, eosinophil and lymphocytes in the BAL of WT and *Ptch1^+/−^* mice with HDM exposure compared to their respective control groups (Figure 7C–F). Although not statistically significant, the percentages of eosinophil and lymphocyte counts were reduced in HDM-exposed *Ptch1^+/−^* mice compared to HDM-exposed WT mice (Figure 7E,F). More strikingly, we showed that HDM-exposed WT mice induced Il-4, Il-5, Il-6, Il-1β, Il-10, Mip3a/Ccl20 and Il12p70 serum protein expression compared to PBS-exposed WT mice, whereas HDM-exposed *Ptch1^+/−^* mice failed to induce these cytokines except for Il12p70 when compared to PBS-exposed *Ptch1^+/−^* mice (Figure 8A–G). Collectively, our data showed *Ptch1* plays an important role in the regulation of a subset of Th1, Th2 and innate immune response-related cytokine expression in HDM-exposed mice.

## 4. Discussion

Our study results collectively demonstrate that TGFβ1-induced increase in COL1A1 expression in airway epithelium is partially mediated through the PTCH1-SMO-GLI axis of the hedgehog signaling pathway. Primary human airway epithelial cells from asthmatic and non-asthmatic cells differentiated in ALI showed significant upregulation in hedgehog-related genes at day 20 compared to day 1 post-ALI. Our silencing and pharmacologic interventional experiments revealed a direct mechanistic link between hedgehog signaling and collagen expression in human airway epithelial cells. We previously have shown that mice haplodeficient in Ptch1 have reduced PTCH1 protein in the airway epithelium compared to those in WT mice [15]. Here, we have confirmed our in vitro findings in these *Ptch1*^+/−^ mice and also showed that they were partially protected from HDM-induced induction of type 2 inflammatory cytokines and airway tissue collagen expression. In summary, we conclude that hedgehog signaling plays an important role in the regulation of sub-epithelial collagen expression, and reveals a new potential therapeutic target for patients with chronic airway diseases such as asthma.

Hedgehog signaling has been reported to be implicated in the pathogenesis of asthma. Zou and colleagues showed that treatment of human bronchial epithelial cell line (16HBE) with Gli1 siRNA or cyclopamine attenuated HDM+TGFβ1-induced increases in COL1 and GLI1 protein [37]. We complemented their findings by showing that SMO siRNA, cyclopamine and GANT61 (inhibitor or GLi1/2) also attenuated TGFb1-induced increase in Col1 expression in 1HAE cell line and ALI culture. In a mouse model of allergic asthma, acute HDM exposure significantly increased TGFβ1 protein in the BAL fluid, decreased expression of CDH1 and occludin in the airway epithelium, and increased expression of VIM, αSMA and pro-collagen 1 in the mesenchyme [16]. In another study, SHH protein in the BAL fluid was up-regulated in children with asthma compared to age-matched non-asthmatic subjects, and mice challenged with ovalbumin were also shown to have elevated *Shh* gene and protein expression in the lungs compared to control mice [15]. It has also been shown that IL4 or IL13 individually induced SHH and GLI activity in human bronchial epithelial cell lines and in vivo in airway epithelium [19]. In the same study, the authors showed that a neutralizing antibody against SHH attenuated ovalbumin-induced increases in macrophage and eosinophil counts in the BAL fluid [15]. However, in the present study, we observed no significant differences in the differential cell counts between HDM-exposed WT and *Ptch1*^+/−^ mice. This discrepancy may be due to the remaining functional copy in *Ptch1*^+/−^ mice providing residual SHH-PTCH1 signaling through the hedgehog pathway. Consistent with data reported by Gregory and colleagues [17], we found that HDM exposure increased Th2-specific Il-4 and Il-5 protein, and some non-specific cytokines such as IL-6, Il-1β, Il-10, Mip3a/Ccl20 and Il12p70 protein. This response was significantly attenuated in the *Ptch1* haplodeficient mice. We also found that while HDM increased collagen expression and Ki67+ cells in the airways, *Ptch1* haplodeficient mice were partially protected against these remodeling changes related to HDM. 

In the context of lung fibrosis, hedgehog signaling has also been implicated in the control of TGFβ1-dependent myofibroblast differentiation in patients with Idiopathic Pulmonary Fibrosis (IPF) [38,39,40]. In line with findings in asthmatic patients, SHH is also increased in the BAL fluid of patients with IPF compared to non-IPF subjects [38]. Hedgehog signaling antagonists attenuate TGFβ1-induced increases in COL1A1, fibronectin and αSMA in both control and IPF fibroblasts [38]. Miller and colleagues demonstrated that αSMA was decreased in the lungs of *Shh*^+/−^ mice and absent in *Shh*^−/−^ mice, indicating that differentiation and/or survival of lung-specific smooth muscle precursors is dependent on SHH [41]. Targeting hedgehog signaling proteins (GLI1/2) with GANT61 reduces bleomycin-induced increases in soluble collagen content in a bleomycin-driven mouse model of lung fibrosis [42]. Consistent with this report, we showed that GANT61 significantly reduced TGFβ-induced increases in *COL1A1* expression in vitro.

There are several important limitations to our study. While we showed in vitro and in vivo that deficiency in *PTCH1* reduces airway tissue remodeling, it is possible that other factors, including oxidative stress, local inflammatory response and microbial interactions, may synergistically activate the hedgehog signaling response involving potential cross-talk with other pathways in the airway tissues of asthmatics. Second, the use of GANT61 resulted in a greater reduction in *COL1A1* and *MMP2* gene expression compared to the use of cyclopamine. One possibility for this observed difference is that GANT61 directly inhibits the transcription factor family of GLI protein, whereas SMO is upstream of GLI and may have multiple modes of signaling [43] that requires further investigations. Third, although we showed the involvement of hedgehog signaling in TGFβ-induced increase in COL1A1 expression in airway epithelial cell cultures, the precise cellular source for the increased collagen deposition in the sub-epithelial compartment in vivo requires further investigation. We regret not having phenotyped the differentiated ALI cultures for the various cell types, as the results shown in Figure 4 and Figure 5 were conducted on unsorted bulk cells from the ALI cultures with different treatments. Xu and colleagues reported that SHH directly induces MUC5AC and SPDEF, and reduces FOXA2 expression in human bronchial epithelial cell line [15], suggesting that hedgehog signaling can promote the conversion of epithelial cells to goblet cells.

Finally, modulating the hedgehog pathway did not reverse the loss of CDH1 expression in the presence of TGFβ stimulation, suggesting that a partial process of epithelial mesenchymal transition may still occur.

In summary, we showed that canonical hedgehog signaling is related to the induction of matrix-related gene expression in our in vitro and in vivo models. This work provides important mechanistic insights into asthma pathophysiology by demonstrating crosstalk between hedgehog and TGFβ signaling, the latter having been implicated to induce EMT in the context of airway tissue remodeling. Our data reveal the hedgehog signaling pathway as a potential target of therapeutic intervention in reducing the synthesis and deposition of excess matrix materials in patients with airway diseases.

## Figures and Tables

**Figure 1 cells-11-03016-f001:**
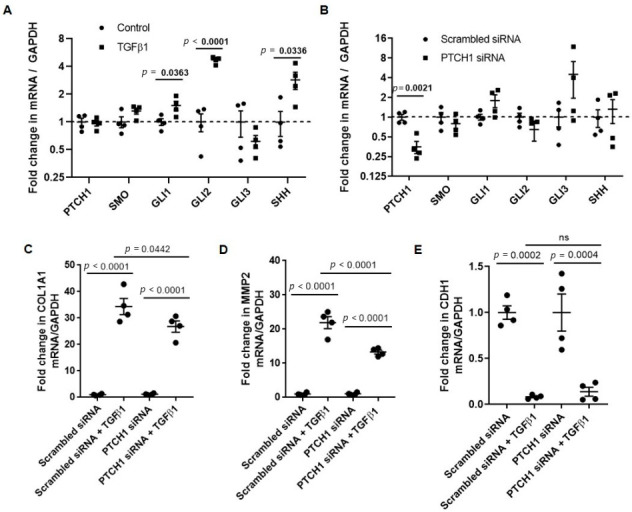
*PTCH1* gene silencing attenuates TGFβ1-induced increase in *COL1A1* and *MMP2* gene expression in 1HAE_o_ cells. Fold change in *PTCH1*, *SMO*, *GLI1*, *GLI2, GLI3* and *SHH* gene expression normalized to *GAPDH* in 1HAEo cells treated with (**A**) TGFβ1 compared to untreated controls and (**B**) PTCH1 siRNA compared to scrambled siRNA were shown. Fold changes in (**C**) *COL1A1*, (**D**) *MMP2,* (**E**) and *CDH1* gene expression normalized to *GAPDH* were assessed in cells treated with scrambled or PTCH1 siRNA in the absence or presence of TGFβ1 treatment. ns = not statistically significant. Values were expressed as mean ± SEM (*n* = 4 independent experiments conducted at different passages). Parametric *t*-test was used in each gene in panel (**A**). One-way analysis of variance with Bonferroni’s multiple comparisons test was used in panels (**B**–**D**).

**Figure 2 cells-11-03016-f002:**
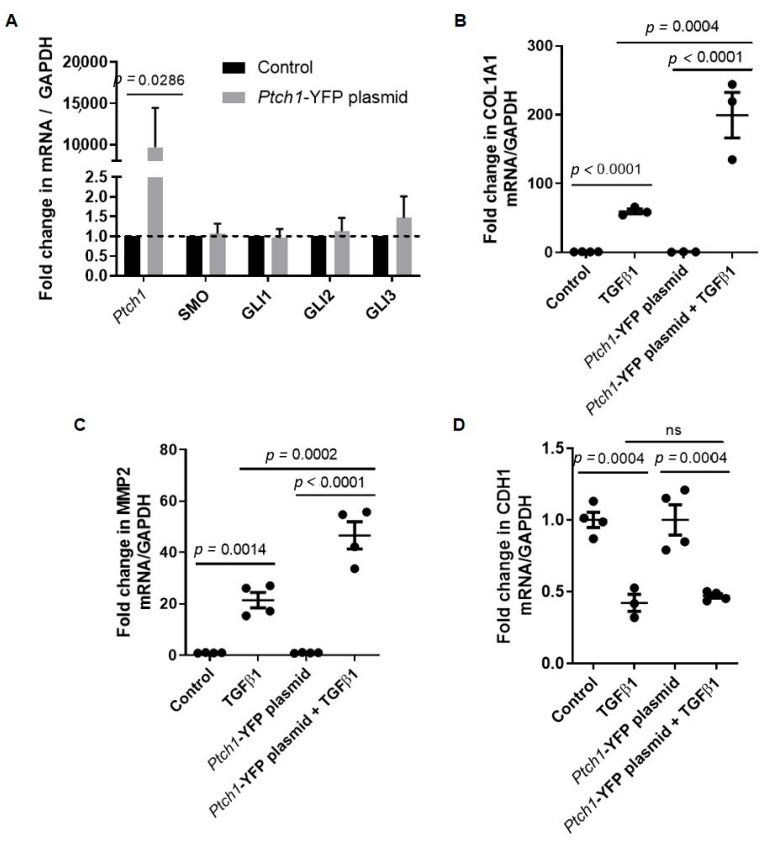
*Ptch1* gene over-expression augments TGFβ1-induced increase in *COL1A1* and *MMP2* gene expression in 1HAE_o_ cells. (**A**) Fold change in *PTCH1*, *SMO*, *GLI1*, *GLI2* and *GLI3* gene expression normalized to *GAPDH* in 1HAEo cells treated with *Ptch1-YFP* gene-containing plasmid compared to untreated control were shown. Fold change in (**B**) *COL1A1*, (**C**) *MMP2* and (**D**) *CDH1* gene expression normalized to *GAPDH* were assessed in cells treated with *Ptch1-YFP* gene-containing plasmid in the absence or presence of TGFβ1 treatment. ns = not statistically significant. Values were expressed as mean ± SEM (N = 3–4 independent experiments). Parametric *t*-test or Mann–Whitney test was used in each gene in panel (**A**). One-way analysis of variance with Bonferroni’s multiple comparisons test was used in panels (**B**–**D**).

**Figure 3 cells-11-03016-f003:**
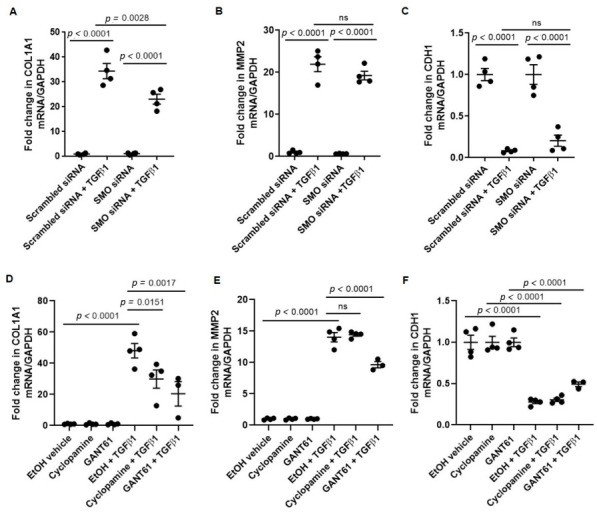
Smoothened (SMO) and GLI1/2 antagonists reduce TGFβ1-induced *COL1A1* gene expression in 1HAE_o_ cells**.** Fold change in (**A**) *COL1A1*, (**B**) *MMP2* and (**C**) *CDH1* gene expression normalized to *GAPDH* were assessed in cells treated with scrambled or SMO siRNA in the absence or presence of TGFβ1 treatment. Fold change in (**D**) *COL1A1*, (**E**) *MMP2* and (**F**) *CDH1* gene expression normalized to *GAPDH* were assessed in cells treated with EtOH vehicle, cyclopamine (SMO inhibitor) or GANT61 (GLI1/2 inhibitor) in the absence or presence of TGFβ1 treatment. ns = not statistically significant. Values were expressed as mean ± SEM (*n* = 3–4 independent experiments). One-way analysis of variance with Bonferroni’s multiple comparisons test was used in panels (**D**–**F**).

**Figure 4 cells-11-03016-f004:**
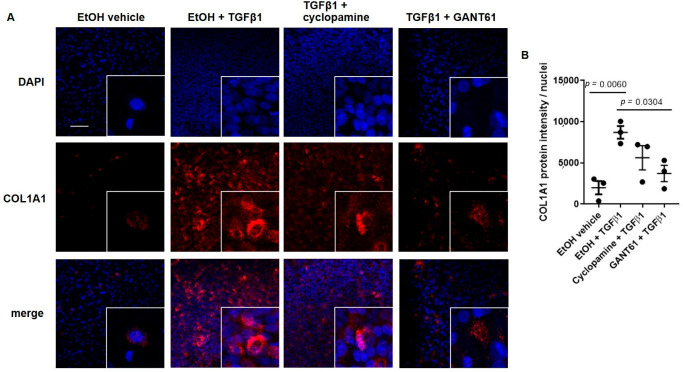
GANT61 attenuates TGFβ1-induced COL1A1 protein expression in primary normal airway epithelial cells differentiated in air–liquid interface (ALI). (**A**) Representative immunofluorescence images from a primary epithelial cell donor differentiated in ALI treated with ethanol (EtOH) vehicle, EtOH + TGFβ1, TGFβ1 + cyclopamine, and TGFβ1 + GANT61 were stained with COL1A1 antibody and counterstained with DAPI. (**B**) COL1A1 protein intensity was normalized to nuclei count. Scale bar = 50 µm. Values were expressed as mean ± SEM (*n* = 3 independent donors were used). One-way analysis of variance with Bonferroni’s multiple comparisons test was used in panel (**B**).

**Figure 5 cells-11-03016-f005:**
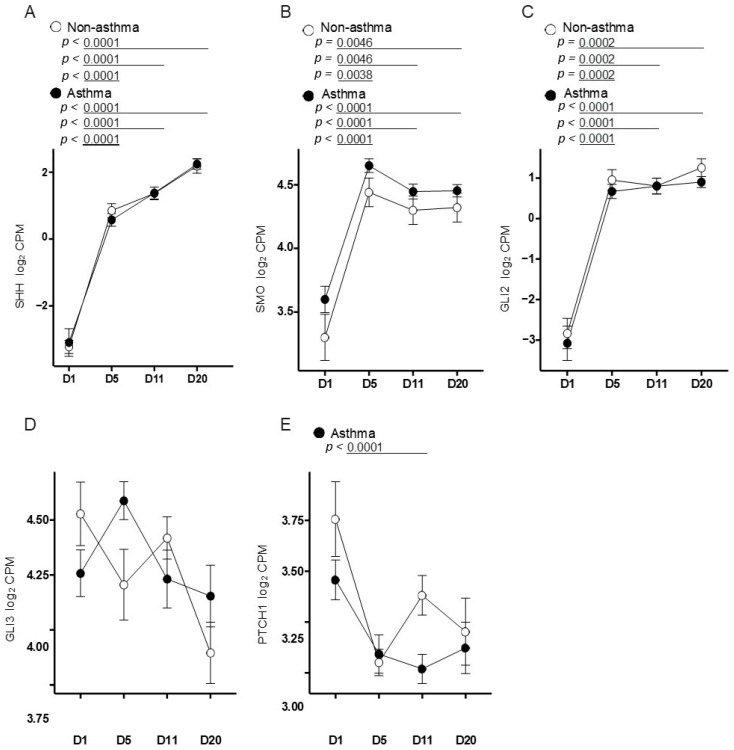
Hedgehog-signaling-related gene expression in differentiated air-liquid interface (ALI) cultures of primary airway epithelial cells. (**A**) *SHH*, (**B**) *SMO*, (**C**) *GLI1*, (**C**) *GLI2*, (**D**) *GLI3* and (**E**) *PTCH1* gene expression from non-asthmatics (*n* = 5) and asthmatics (*n* = 10) primary airway epithelial cells (PAECs) cultured for 1-, 5-, 11- or 20 days post-ALI were expressed as Reads Per Kilobase of transcript, per Million mapped reads (RPKM). One-way ANOVA with Dunnett’s multiple comparisons test was used in panels (**A**–**E**).

**Figure 6 cells-11-03016-f006:**
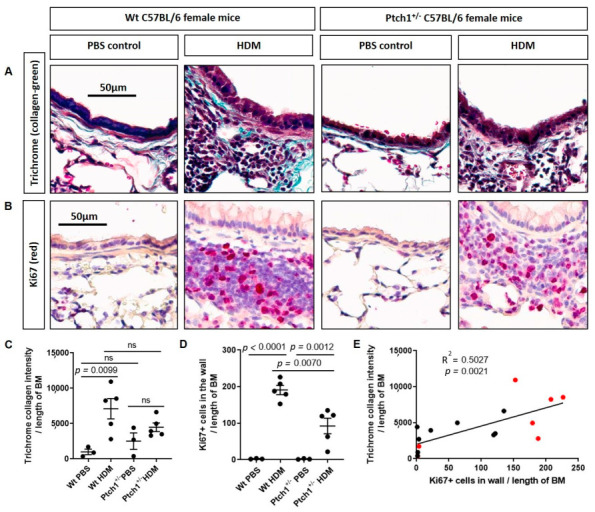
*Ptch1*^+/−^ mice were partially protected from HDM-induced increase in total collagen expression and Ki67+ cells in the sub-epithelial walls compared to WT mice. Paraffin−embedded mouse lung tissues from C57BL/6 female WT PBS−treated (*n* = 3), WT HDM−treated (*n* = 5), *Ptch1*^+/−^ WT PBS-treated (*n* = 3) and *Ptch1*^+/−^ HDM-treated (*n* = 5) mice after PBS or HDM exposures were stained with (**A**) Trichrome and (**B**) Ki67 antibody. Sub−epithelial (**C**) collagen expression (light green) and (**D**) Ki67+ cells (red) were normalized to the length of the basement membrane. (**E**) The correlation between sub−epithelial collagen expression and Ki67+ cells is shown. One-way analysis of variance with Bonferroni’s multiple comparisons test was used in panels (**C**,**D**). Linear regression analyses were used in panel (**E**). Red dot = WT PBS and HDM−treated mice, black dot = *Ptch1*^+/−^ PBS and HDM−treated mice.

**Figure 7 cells-11-03016-f007:**
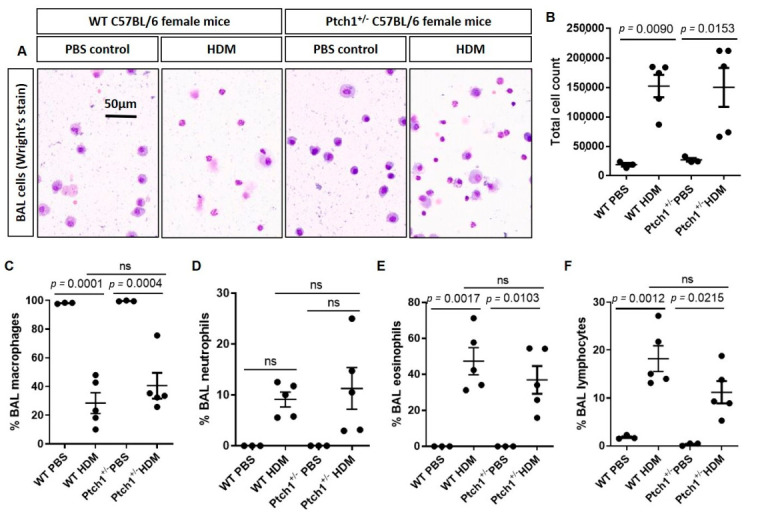
Differential cell counts from broncho−alveolar lavage (BAL) fluid were profiled in WT and *Ptch1*^+/−^ mice with or without HDM exposure. (**A**) Representative images of cells stained with Wright’s stain from BAL of WT and *Ptch1*^+/−^ mice with or without HDM exposure were shown. Scale bar = 50 µm. (**B**) Total cell count, (**C**) % macrophages, (**D**) % neutrophils, (**E**) % eosinophils and (**F**) % lymphocytes from BAL were quantified. One-way analysis of variance with Bonferroni’s multiple comparisons test was used in panels (**B**–**F**).

**Figure 8 cells-11-03016-f008:**
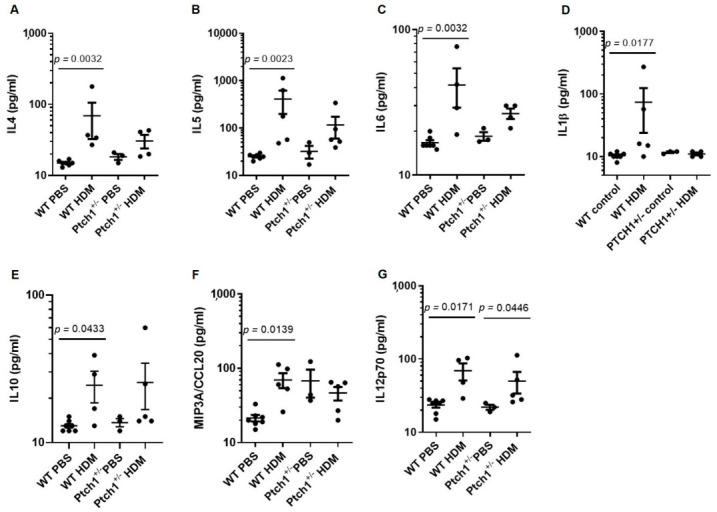
Serum cytokines were profiled in WT and *Ptch1*^+/−^ mice with or without HDM exposure. (**A**) Il-4, (**B**) Il-5, (**C**) Il-6, (**D**) Il-1β, (**E**) Il-10, (**F**) Mip3a/Ccl20 and (**G**) Il12p70 protein measured in the serum of WT and *Ptch1*^+/−^ mice with or without HDM exposure were shown. Kruskal−Wallis with Dunn’s multiple comparisons tests were used in panels (**A**–**G**).

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
