# Peer review of "Hedgehog Signaling as a Therapeutic Target for Airway Remodeling and Inflammation in Allergic Asthma"

_cells, 2022, doi:10.3390/cells11193016_

Round 1
Reviewer 1 Report (Previous Reviewer 1)
Thank you to the authors for carefully considering the previous critiques and for addressing the comments made.
There are no further concerns.
Reviewer 2 Report (Previous Reviewer 2)
Thank you for the responses to my questions. I have no further questions.
This manuscript is a resubmission of an earlier submission. The following is a list of the peer review reports and author responses from that submission.
Round 1
Reviewer 1 Report
Cells, Article by Tam et. al. on Hedgehog signaling.
The manuscript draft presented by Tam et al. seeks mechanistic insight into the GWAS association of PTCH1 (patched homolog 1) with lung function in the general population. Sonic hedgehog (SHH) is a ligand for PTCH1 and it is also up-regulated in asthma, suggesting that the signaling pathway involving these molecules is important for normal airway physiology and remodeling. Gene silencing, over-expresson and pharmacologic inhibition were conducted in human airway epithelial cells. TGFB1 was also utilized in manipulation cultures. In vivo HDM challenge models in Ptch1 +/- mice were also utilized. A role for this signaling pathway as a therapeutic target for allergic airways disease is suggested by the findings.
Major Comments:
Line 102; were the over-expression studies conducted as a “transient” transfection. If so, how quickly was the transgene lost?
Figure 1A, importance? Why show these genes, but then not mention effect of PTCH1 manipulation on these genes? Meaning is not clear here. Why not show all the data in this figure as you do in 1B (rather than just bars reflecting means and SEM?).
Figure 1: What is meant by N=4 experiments? Conducted on different days? Conducted using passaged cells?
GAPDH as a reference for qPCR: please confirm that expression levels were not altered with treatment. You can look at the cross points.
What is the control in Figure 2, over-expression studies? The plasmid w/o the transgene? YFP expression only?
Purpose of Figure 2A? As with Figure 1A, what happens to these genes after manipulation with PTCH1 over-expression?
Not sure differences discussed in Figure 4C are real. RPKM values for this gene are very low. Are you down in the “noise?” Is Figure 4 data from a published dataset? If so, this should be clearly stated so that no one thinks this is “new data” that couldn’t be explored without obtaining the new data associated with this paper.
Not sure how data from Figure 4 contributes to the conclusion on line 283-284.
Not sure why text description of figure 5 ignores Cyclopamine. Figure 5 seems better suited before Figure 4 as it seems associated with gene expression data in Figure 3.
Figure 6C. The statistic between columns 2 and 4 is not given. Is there more collagen in Ptch1 +/- mice at baseline?
Lines 329-331… can you say percentages were reduced?
Figure 8; likely there is no difference in cytokine levels between WT HDM and Ptch+/- HDM treatment. Are you sure you have not overinterpreted your findings in the text. One outlier animal in WT may be driving this. Is line 374-375 true?
Minor Comments:
Line 53, reference with hyphen (5-) is odd.
Lines 59-61, formulation of thoughts is odd. GLI factors defined twice? Perhaps can be reworked.
Lines 65-69; can you comment on specificity of SMO and GLI to the hedgehog pathway? Would the inhibitors we expected to disrupt signaling in pathways besides SHH?
Lines 74-76; Any evidence in airway epithelial cells for this type of crosstalk? Since this paper focuses on epithelial cell models, SHH signaling in these cells might be more relevant (as opposed to eosinophils?)
Line 164; Mice, which are homozygous…. Should be “Mice homozygous…”
Line 207, grammar seems off.
Figures 1-2, did you think all three genes should act the same?
Figure 1D, 2D; it is the statistic between columns 2 and 4 that is most important here, yes? Shown in B, C, but not D.
Line 335, were all these cytokines Type 2?
Figure 7, interesting comparison is between genotype after HDM. Assume not significant, but that is not shown on graphs. Would be useful to mention how many days after HDM treatment these cell counts were taken.
Discussion could be cut by at least 1/3. Paragraph lines 378-397 does not seem particularly relevant.
Reviewer 2 Report
The manuscript by Tam and colleagues describes the involvement of hedgehog signaling in TGFbeta-induced responses in a cell line and primary cell culture and validates some results in a murine model of HDM exposure. The authors demonstrate a link between TGFbeta-induced collagen 1 production and hedgehog signaling. The manuscript has an elegant and structured approach and is well written, but the results could be extended to increase the novelty of the results.
A publication by Zou et al in 2019 in Mol. Med. Rep. (PMID: 31702025), already describes the link between TGFbeta (combined with HDM) and hedgehog signaling in TGFb-induced increase in collagen 1. This paper is not discussed in the manuscript and it would be good to include it and explain more clearly how the results in the present manuscript provide additional insights to these results.
Why was specifically TGFbeta chosen in the study and not in combination with for example IL-13, that also induces asthma-related epithelial remodeling? Using only TGFbeta makes the link to asthma a bit arbitrary as it is also involved in fibrosis- or COPD-related remodeling. Or is it already established that hedgehog signaling is a contributing factor in IL-13-induced airway epithelial cell remodeling?
It was interesting to see that hedgehog-related proteins such as SHH and SMO develop in culture over differentiation time, suggesting they are expressed by luminal epithelial cell-types. Could it therefore also be that the increase in hedgehog signaling found in the GWAS is a consequence of the remodeling of the airways by promoting a cell-type (e.g. goblet cells) that expresses SHH/SMO etc., rather than a driver per se? Can the authors check in their cell cultures whether the changes in collagen I are related to changes in cell types by TGFbeta and/or GANT61 exposure, rather than a direct effect on the signaling?
The methods section describes RNA isolation for RNA sequencing from the primary bronchial epithelial cell cultures, but I do not find and reports on results from RNAseq experiments.
Can the authors verify that the right statistical tests were used, possibly via a statistician? For example in Fig. 5 there are two factors: time and asthma/non-asthma. In my view a two-way ANOVA should be used here, not one way. Also statistical significance seems really high with only N=4, for example in the comparison between scrambled siRNA + TGFb1 and SMO siRNA + TGFb1 in Fig. 3A, or in Fig. 2C between TGFb1 and plasmid + TGFb1, can the authors verify that the right test was used?
Can the authors include scale bars in the imaging pictures in Figure 5A. Why is there a difference in cell density (based on the DAPI staining) between the different conditions?
On lines 307 it is stated that HDM-exposed Ptch+/- mice show reduced levels of collagen I and KI67, but this is not shown for collagen I in Fig. 6C. Please formulate this a bit more careful.
Reviewer 3 Report
I have carefully read the article entitled "Hedgehog Signaling as Therapeutic Target on Airway Remod-2 eling and Inflammation in Allergic Asthma' by Tam et al. It is an interesting and novel study which has good results and conclusions.